# miR-196a Upregulation Contributes to Gefitinib Resistance through Inhibiting GLTP Expression

**DOI:** 10.3390/ijms23031785

**Published:** 2022-02-04

**Authors:** Bing-Jie Liu, Fang-Fang Li, Yun-Xia Xie, Chong-Yuan Fan, Wen-Jing Liu, Jian-Ge Qiu, Bing-Hua Jiang

**Affiliations:** 1Academy of Medical Sciences, Zhengzhou University, Zhengzhou 450052, China; bingjieliu@zzu.edu.cn (B.-J.L.); 15238331082@163.com (F.-F.L.); xieyunxiaxyx@163.com (Y.-X.X.); f17637501529@163.com (C.-Y.F.); 2Department of Pathology, Anatomy and Cell Biology, Thomas Jefferson University, Philadelphia, PA 19107, USA; zlyyliuwenjing1146@zzu.edu.cn

**Keywords:** non-small cell lung cancer, gefitinib, miR-196a, NRF2, GLTP

## Abstract

Tyrosine kinase inhibitor (TKI) therapy has greatly improved lung cancer survival in patients with epidermal growth factor receptor (EGFR) mutations. However, the development of TKI-acquired resistance is the major problem to be overcome. In this study, we found that miR-196a expression was greatly induced in gefitinib-resistant lung cancer cells. To understand the role and mechanism of miR-196a in TKI resistance, we found that miR-196a-forced expression alone increased cell resistance to gefitinib treatment in vitro and in vivo by inducing cell proliferation and inhibiting cell apoptosis. We identified the transcription factor nuclear factor erythroid 2-related factor 2 (NRF2) bound to the promoter region of miR-196a and induced miR-196a expression at the transcriptional level. NRF2-forced expression also significantly increased expression levels of miR-196a, and was an upstream inducer of miR-196a to mediate gefitinib resistance. We also found that glycolipid transfer protein (GLTP) was a functional direct target of miR-196a, and downregulation of GLTP by miR-196a was responsible for gefitinib resistance. GLTP overexpression alone was sufficient to increase the sensitivity of lung cancer cells to gefitinib treatment. Our studies identified a new role and mechanism of NRF2/miR-196a/GLTP pathway in TKI resistance and lung tumor development, which may be used as a new biomarker (s) for TKI resistance or as a new therapeutic target in the future.

## 1. Introduction

Lung cancer is one of the most serious human malignant tumors. The morbidity and mortality of lung cancer are much higher than most malignant cancers [1]. Non-small cell lung cancer (NSCLC) accounts for about 80% of all lung cancer cases, and approximately 75% of patients are already in the advanced stages when they are first diagnosed. Surgery, chemotherapy and radiotherapy are the main therapeutic methods for lung cancer treatments; neoadjuvant and immunotherapy for advanced or metastatic lung cancer. However, the 5-year survival rate of lung cancer patients is still very low [2,3]. With the development of gene sequencing technology, gene driven mutations have been studied, and the promising treatments of targeted drugs bring hope to patients with special gene mutations. Targeted therapy has achieved good outcomes in the treatment of Stage IV lung cancer [4,5].

Epidermal growth factor receptor (EGFR) gene mutation occurs in about 10–50% of NSCLC tumors depending on the population [6,7]. EGFR exon 19 deletions and exon 21 L858R point mutations account for about 85% of somatic EGFR alterations and predict sensitivity to EGFR-tyrosine kinase inhibitors (EGFR-TKIs) [2]. EGFR-TKIs may significantly improve the median progression-free survival (PFS) of patients carrying specific site mutations or deletions of EGFR [8]. Gefitinib is the first generation of EGFR-TKIs that was one of the first TKI drugs approved by the FDA for treatment of locally advanced or metastatic non-small cell lung cancer patients [9]. Up to now, the National Comprehensive Cancer Network have listed more than five EGFR-TKIs, including the third generation of EGFR-TKIs, osimertinib [10]. Although EGFR-TKIs have an excellent initial clinical therapeutic effect, the intrinsic resistance and acquired resistance to EGFR-TKIs pose a major barrier to the widespread use of EGFR-TKIs in clinical lung cancer treatment. About 20%–30% of NSCLC patients harboring activating EGFR mutations have no good initial clinical responses to EGFR-TKIs, but almost all patients finally develop acquired resistance after 10–12 months of treatment [11,12]. Therefore, it is important to understand new mechanisms of EGFR-TKI-acquired resistance to explore new approaches for lung cancer therapy in the future.

MicroRNAs (miRNAs) are non-coding RNAs with a sequence of 20–24 nucleotides. MicroRNAs regulate cancer development and/or drug resistance by targeting mRNAs. In recent years, numerous studies have shown that microRNAs are involved in the regulation of tumor growth, metastasis and angiogenesis in different cancers [13,14,15,16]. Our previous study showed that miR-199 and miR-497 inhibited tumor growth and attenuated chemoresistance in glioma and lung cancers [17,18]. We also found that miR-196a expression was induced by estrogen and promoted tumor growth and metastasis [19]. miR-196a levels were highly expressed in gastric cancer and glioblastoma, which promoted cell proliferation and migration [20,21]. Serum miR-196a levels were significantly increased in lung cancer patients and miR-196a regulated the proliferation, migration and invasion of NSCLC cells by downregulation of homeobox protein HoxA5 [22,23]. However, the role of miR-196a in TKI therapy resistance in lung cancer is not known yet.

In this study, we expect to address the following questions: (1) whether miR-196a expression is induced in gefitinib-resistant lung cancer cells; (2) whether higher expression levels of miR-196a are required for gefitinib resistance; (3) the regulation mechanism of miR-196a upregulation and downstream target gene (s) of miR-196a that are/is important in gefitinib resistance; (4) whether higher miR-196a expression levels are associated with gefitinib resistance in vivo.

## 2. Results

### 2.1. miR-196a Expression Levels Were Greatly Upregulated in Gefitinib-Resistant Lung Cancer Cells, and Its Levels Were Positively Correlated with Poor Overall Survival in Lung Cancer Patients

To test whether miR-196a was involved in gefitinib resistance, we analyzed the database from gene expression omnibus (GEO) and found that miR-196a levels were significantly higher in the lung cancer cells from gefitinib treatment-resistant cells than those from gefitinib sensitive cells (Figure 1a,b). We detected expression levels of miR-196a in PC-9 and gefitinib-resistant PC-9-G cell lines, the results showed that the expression levels of miR-196a were significantly higher in gefitinib-resistant PC-9-G cells than those in PC-9 cells (Figure 1c). The correlation of the patient (without gefitinib inducement) survival and miR-196a levels in lung adenocarcinoma (LUAD) was analyzed using The Cancer Genome Atlas (TCGA) database; the result showed that patients with higher miR-196a expression levels had significantly lower overall survival rates (Appendix A). Taken together, these results indicated that higher expression levels of miR-196a in lung cancer cells were associated with gefitinib resistance, which indicated that miR-196a may be a potential therapeutic target to overcome gefitinib resistance, which is to be investigated in the future.

We further tested the mRNA expression level of miR-196a in 17 pairs of lung cancer and adjacent tissues and found that miR-196a levels were significantly upregulated in tumor tissues compared to their corresponding adjacent tissues (Figure 1d and Appendix A). To confirm the expression pattern of miR-196a in lung cancer, we obtained miR-196a expression levels in 512 different lung cancer tissues and 20 normal samples from TCGA database; these showed that the expression levels of miR-196a were much higher in lung cancer tumor tissues than in normal tissues (Appendix A). All of these results suggested that miR-196a might contribute to the occurrence and development of lung cancer as well as gefitinib resistance.

### 2.2. Forced Expression of miR-196a-Induced Gefitinib Resistance and Cell Proliferation, and Inhibited Cell Apoptosis That Was Important in Cell Resistance to Gefitinib Treatment

To test the direct role of miR-196a in gefitinib resistance, we constructed miR-196a over-expression cell lines (miR-196a) and control cell lines (miR-NC) in PC-9 cells (Appendix A). The cells mentioned above were treated with different concentrations of gefitinib for 48 h; the CCK8 assay showed that miR-196a overexpression enhanced the resistance to gefitinib treatment in PC-9 cells compared to the miR-NC group (Figure 2a). The results indicated that miR-196a may be involved in the gefitinib resistance of lung cancer cells. To study the function of miR-196a in the gefitinib resistance of lung cancer, the CCK8 assay suggested that miR-196a overexpression promoted the cell proliferation (Figure 2b). Upregulation of miR-196a promoted cell proliferation by increasing the S phase of the cell cycle (Figure 2c,d). Moreover, our studies showed that miR-196a inhibited cell apoptosis in PC-9 cells (Figure 2e,f). The miR-196a overexpression cells expressed higher levels of BCL-2 and lower levels of BAX than the miR-NC cells (Figure 2g). These results suggested that miR-196a boosted lung cancer cells’ resistance to gefitinib treatment by promoting cell proliferation and inhibiting cell apoptosis.

### 2.3. NRF2 Expression Levels Were Increased in Gefitinib-Resistant Cells and NRF2 Induced miR-196a Expression at Transcriptional Level

To further identify the upstream molecule (s) leading to miR-196a upregulation, we predicted potential regulatory transcription factors of miR-196a via the Jasper website, and our analysis showed that the transcription factor NRF2 encoded by *NFE2L2* had three potential binding sites in the promoter region of miR-196a (Figure 3a). We also analyzed the correlation of Yin and Yang 1 Protein (YY1) transcription factor, but the NRF2 obtained the higher score (Appendix A). We speculated that NRF2 may play a role in the transcriptional regulation of miR-196a. NRF2 acted as a transcription regulator and the bases of the functional region were shown (Figure 3b). Moreover, we confirmed that the expression levels of NRF2 were higher in gefitinib-resistant PC-9-G cells than in PC-9 cells (Figure 3c). We predicted that the high expression of NRF2 promoted the expression of miR-196a through transcriptional regulation in PC-9-G cells. To confirm the regulating function of NRF2 for miR-196a expression, we constructed NRF2 overexpression cells in PC-9 cells; Western blot assay verified that the stable transformed cell line of NRF2 was successfully constructed (Figure 3d). In the NRF2 overexpression stable cell line, we measured the mRNA levels of miR-196a and found that NRF2 overexpression significantly increased the expression levels of miR-196a (Figure 3e). To further determine the interaction relationship between miR-196a and NRF2, the luciferase reporter assay confirmed that NRF2 directly bound to the promoter region of miR-196a and increased the promoter activities of miR-196a (Figure 3f). YY1 had potential binding sites in the promoter region of miR-196a, but the score was not high. The luciferase reporter experiment suggested that YY1 has no direct role in regulating on miR-196a (Appendix A). These results suggested that high expression levels of miR-196a in lung cancer cells were directly induced by NRF2 through the transcriptional activation.

### 2.4. GLTP Was a Direct Functional Target of miR-196a, and GLTP Suppression Was Important for the Resistance of Cells to Gefitinib Treatment

MicroRNAs usually regulate tumor biological function through their downstream target genes. To further explore the mechanism of miR-196a in mediating gefitinib resistance in lung cancer cells, we predicted the downstream target genes of miR-196a via four databases included in TargetScan, miRana, miR-walk and miR-TCD, and we obtained 79 overlapping candidate target genes for miR-196a in the Venn diagram (Figure 4a). According to relevant literature research and preliminary detection, we screened out GLTP, PTPRG and BIRC6 from 79 candidate target genes for further confirmation. Moreover, we analyzed the relative expression levels of these 79 predicted targets based on fold changes and abundance in the transcriptome data of gefitinib resistance in HCC4006 cells, and further selected three top target genes. The results showed that GLTP levels were the lowest expression gene in gefitinib resistance cells (Appendix A). In the drug-resistant cells, the correlation between three potential target genes and miR-196a was analyzed, and the analysis showed that GLTP was significantly negatively correlated with miR-196a (Figure 4b). The data of HCC4006 cells and HCC4006 gefitinib resistance cells indicated that GLTP may be the direct target gene of miR-196a. We also detected the mRNA expression levels of the above three genes in our gefitinib-resistant cell lines and paired tumor tissues. Our results showed that the three candidate genes were all downregulated in PC-9-G cells (Figure 4c and Appendix A) and tumor tissues (Appendix A). These results indicated that all three target genes were probably involved in tumor drug resistance or promoted the occurrence of lung cancer, but only GLTP levels were decreased significantly with *p* < 0.01 in miR-196a overexpression cells (Figure 4d). However, forced expression of miR-196a did not decrease mRNA expression levels of BIRC6 and PTPRG, two other targets (Appendix A). We speculated that GLTP is the direct and functional target gene of miR-196a. Further results showed that the protein expression levels of GLTP were significantly lower in PC-9-G cells and miR-196a overexpression cells than control groups (Figure 4e). To further determine whether GLTP was a direct target of miR-196a, we designed GLTP wide type (GLTP-WT) 3′-UTR and mutant sequences (GLTP-Mut) for luciferase reporter assay (Figure 4f). miR-196a combined with the 3′-UTR of GLTP-WT and reduced the transcription activity of GLTP, but not in the GLTP-Mut group (Figure 4g). From the GEO database, we also found that the expression levels of GLTP were extremely low in the gefitinib resistance cells (Figure 4h), and the expression levels of GLTP were decreased with the increase in tumor grade (Figure 4i). TCGA database analysis showed that the lung cancer patients with lower expression levels of GLTP had a shorter time of survival than those with higher levels of GLTP (Figure 4j). These results indicated that GLTP played an important role in the malignant stages, which was suggested as a functional target of miR-196a to regulate gefitinib resistance in lung adenocarcinoma.

To explore the role of GLTP in the gefitinib resistance of NSCLC, we knocked down GLTP via shRNA in the PC-9 cell and overexpressed GLTP in the PC-9-G cell (Figure 5a–c). Cell viability was detected in GLTP knockdown or overexpression cells that were treated with different concentrations of gefitinib. Our results suggested that the knockdown of GLTP decreased the sensitivity to gefitinib treatment in PC-9 cells and the opposite result occurred with the ectopic expression of GLTP in PC-9-G cells (Figure 5d,e). The experimental results pointed out that GLTP probably played a regulatory role in the gefitinib resistance of PC-9 cells. Furthermore, we examined the cell apoptosis in GLTP knockdown and the overexpression of stable cell lines, we found that knockdown of GLTP in PC-9 cells inhibited cell apoptosis, especially the early apoptosis rate (Figure 5f,g), GLTP-forced expression accelerated cell apoptosis in PC-9-G cells (Figure 5h,i). Flow cytometry assay results suggested that GLTP knockdown may increase the resistance of PC-9 cells to gefitinib by inhibiting cell apoptosis. Western blot assay also showed that GLTP-forced expression decreased the expression levels of BCL-2 and increased BAX expression in PC-9-G-constructed stable cell lines (Figure 5j). These results indicated that GLTP expression increased sensitivity to gefitinib treatment through inducing cell apoptosis in NSCLC.

### 2.5. GLTP-Forced Expression Inhibited miR-196a-Inducing Gefitinib Resistance

Our studies showed that GLTP played a key regulatory role in the gefitinib resistance of NSCLC. In order to confirm whether miR-196a regulating the gefitinib resistance was mainly through GLTP, we established miR-196a and GLTP co-expression cells. Our results showed that miR-196a overexpression decreased the cell sensitivity to gefitinib treatment in lung cancer and this effect was partly reversed by the upregulation of GLTP (Figure 6a). Our results confirmed that the effect of miR-196a on gefitinib resistance was achieved by targeting GLTP. Flow cytometry assay suggested that GLTP-forced expression also induced cell apoptosis, the opposite effect of regulation by miR-196a (Figure 6c,d). We also detected expression levels of apoptosis-regulating proteins, BCL2 and BAX, and showed that miR-196a induced BCL-2 but inhibited BAX expression; overexpression of GLTP can reverse the effect of miR-196a on apoptosis (Figure 6b). Taken together, our results demonstrated that GLTP was a functional downstream target gene of miR-196a in regulating cell resistance to gefitinib treatment in NSCLC.

### 2.6. miR-196a-Forced Expression Induced Tumor Growth and Inhibited Cell Apoptosis In Vivo, Which May Contribute to Gefitinib Resistance

To further confirm the role of miR-196a in vivo, cell lines with or without miR-196a overexpression were subcutaneously injected into 4-week-old nude mice, and 14 days later, the mice of the two groups were given gefitinib or vehicle by gavage every two days. Ectopic expression of miR-196a promoted tumor growth with gefitinib treatment; seven weeks later, the mice were euthanized, and tumors were taken out for photography and analysis (Figure 7a). The tumor volumes of mice were measured once a week, and our results showed that the tumors in the miR-196a overexpression group were significantly larger than those in the negative control group (Figure 7b). Proliferation marker ki67 expression was detected in the tumor tissues by IHC staining, and the results showed that the expression levels of ki67 were higher in the miR-196a overexpression groups than those in the control groups (Figure 7c,d). The tumorigenesis experiment confirmed that miR-196a promoted tumor growth under the treatment of gefitinib. Furthermore, the BCL-2 levels were upregulated, and BAX levels were downregulated in miR-196a groups compared with the control groups (Figure 7e–h). These in vivo results showed that miR-196a overexpression induced tumor growth and inhibited cell apoptosis, which contributed to gefitinib resistance in NSCLC.

## 3. Discussion

EGFR-TKIs (such as gefitinib) significantly increase the overall survival of lung cancer patients with EGFR mutants. However, about 40% of the patients developed drug resistance after 0.5–1 year of treatment. Up to now, resistance of TKI-targeted therapy has been a major problem in lung cancer therapy. It is necessary to elucidate the new mechanisms of EGFR-TKIs resistance. High expression levels of FOXO3a, BAX or FBXW7 were reported to be associated with EGFR mutation-independent EGFR-TKIs sensitivity, suggesting that targeting these key genes had potential therapeutic value [24,25,26]. Non-coding RNAs may also be important in NSCLC drug resistance [27,28,29]. It has been reported that miR-196a as an oncogene promoted the migration and invasion of lung cancer and osteosarcoma [30,31,32]. miR-196a was also reported to elevate the cell proliferation in different cancer types [32,33,34]. However, the role and mechanism of miR-196a in gefitinib resistance were unknown. In this study, we unveiled a new role of miR-196a in gefitinib resistance in NSCLC.

To further explore the mechanism of miR-196a upregulation in gefitinib resistance, we identified GLTP as a new direct functional target of miR-196a in mediating gefitinib resistance. GLTP is located on Chromosome 12, which encodes a 24KDa protein and mediates glycosphingolipid intermembranes trafficking and regulates glycosphingolipid homeostatic levels under physiological conditions. Most studies showed that GLTP and its superfamily members selected and transferred glycosphingolipids in normal physiological processes [35,36]. GLTP induced cell cycle arrest and necroptosis in certain colon cancer cell lines to effect cell death [37]. Recently, studies indicated that GLTP was involved in the process of the autophagy, inflammation and cell death of cancer cells [38]. In our study, we showed that GLTP may be an important target of miR-196a in promoting gefitinib resistance through cell apoptosis.

To identify the upstream inducer of miR-196a, we found the transcription factor NRF2 binding sites in the promoter region of miR-196a. NRF2 was encoded by the *NFE2L2* gene and involved in the regulation of cell homeostasis, including antioxidant proteins, detoxification enzymes, drug transporters and other proteins [39]. NRF2 was considered to have two sides to its roles, it may play as either an oncogene or a tumor suppressor gene based on different cellular environments. NRF2 activation played a protection role in chemical- and radiation-induced carcinogenesis [40,41]. In recent years, more and more studies have found that NRF2 activation could promote the process of tumor growth [42,43]. In our study, we showed that NRF2 increased the expression of miR-196a, and high expression of miR-196a contributed to the gefitinib resistance. We identified the region of the NRF2 binding sites in the promoter of miR-196a and determined that NRF2 increased the expression levels of miR-196a through transcriptional activation. However, the binding domain of miR-196a and NRF2 remains to be defined, which can be explored in the future.

Our findings provided the first evidence that the NRF2/miR-196a/GLTP pathway was an important mechanism of gefitinib resistance, and miR-196a and GLTP may be used as new biomarkers for gefitinib resistance and clinical therapeutic outcomes of lung cancer. A combination of targeting miR-196a and gefitinib target therapy may be a new promising therapeutic approach to overcome lung cancer TKI resistance and tumor growth.

## 4. Materials and Methods

### 4.1. Human Lung Cancer Specimens

In this study, the human lung cancer tumor tissues and adjacent normal tissues of lung cancer patients were obtained from the tissue bank of the Affiliated Cancer Hospital of Zhengzhou University (Zhengzhou, China). The lung cancer tissues had been collected and stored in the tissue bank for several years. The patients did not receive surgical treatment, radiotherapy, chemotherapy or immunotherapy before surgery. The patient information including names, age and other patient personal information was not known to the investigators.

### 4.2. Cell Culture and Reagents

Lung cancer cell line PC-9 was purchased from ATCC and its corresponding gefitinib-resistant cell line PC-9-G was constructed by increasing concentrations of gefitinib for two years. The paired cell lines were cultured in RPMI-1640 medium supplemented with 10% fetal bovine serum (FBS) (Gibco, Waltham, MA, USA) and 1% penicillin/streptomycin (Gibco, Waltham, MA, USA). All cells were maintained in a 37 °C incubator with 5% carbon dioxide (CO_2_). The gefitinib (MCE, Monmouth Junction, NJ, USA) stock solution was prepared in sterile water at 10 mM and stored at −20 °C.

### 4.3. RNA Extraction and Quantitative Real-Time PCR Analysis

Total RNAs of lung cancer tissues and gefitinib-resistant cell lines were extracted by TRIzol Reagent (Invitrogen, Waltham, MA, USA), and the concentrations of extracted RNAs were detected by Nanodrop (Thermo Scientific, Waltham, MA, USA). The cDNAs were obtained from 1 μg extracted RNAs using HiScript III 1st Strand cDNA Synthesis Kit (Vazyme Biotech, Nanjing, China). The expression levels of miR-196a and its target gene (s) were detected using SYBR Green Master mix PCR kit (Vazyme Biotech, Nanjing, China) by PCR system (ABI, Los Angeles, CA, USA). All qPCR primers used are provided in Appendix A.

### 4.4. Cell Viability Assay

Cells were seeded into 96-well plate (2000–3000 cells/well) with four duplicated wells and cultured for 3, 5 and 7 days. Then 10 μL of CCK-8 solution (Vazyme Biotech, Nanjing, China) per well was added and incubated for 2 h at 37 °C. The absorbance was measured by microplate reader at 450 nm (Molecular Devices, San Jose, CA, USA).

### 4.5. Cell Apoptosis Assay

Cells were seeded into 6-well plates and treated with different concentrations of gefitinib for 48 h. The cells and cultured medium were collected and centrifuged at 1000× *g* for 5 min, then washed with cold PBS and stained with the FITC-Annexin V/.PI Apoptosis Detection Kit (BD Biosciences, Fremont, CA, USA) or the PE-Annexin V/7-ADD Apoptosis Detection Kit (Vazyme Biotech, Nanjing, China). The stained samples were analyzed using flow cytometer (BD Biosciences, Fremont, CA, USA) within 1 h.

### 4.6. Western Blot Analysis

Cells were harvested, and tumor tissues were lysed in RIPA buffer containing a protease inhibitor cocktail (1:100, Beyotime Biotechnology, Shanghai, China). Total proteins were extracted and quantitated by a BCA protein assay kit (Thermo Fisher Scientific, Waltham, MA, USA). Then 10 mg total proteins were separated by 8% SDS-PAGE and subsequently transferred to polyvinylidene difluoride (PVDF) membranes (Millipore, Burlington, MA, USA). The membranes were blocked with 5% non-fat milk in TBST for 1 h at room temperature, then incubated with primary antibodies (anti-GLTP, anti-BCL-2, anti-BXA, 1:1000 from Proteintech, Chicago, IL, USA) at 4 °C overnight. The membranes were incubated with horseradish peroxidase-conjugated goat anti-mouse or anti-rabbit IgG (1:5000, Signalway Antibody, College Park, MD, USA) secondary antibodies for 1 h at room temperature. Protein bands were visualized by ECL Western Blot Substrates (Thermo Fisher Scientific, Waltham, MA, USA).

### 4.7. Transfection

The negative control and miR-196a mimics were purchased from RIBOBIO (Guangzhou, China). Cells were plated into 6-well plates to reach a confluence of 50%-70%. Then, 100 nM miR-196a mimics were transfected into corresponding cells following the instruction of transfection reagents (Roche, Indianapolis, IN, USA) for 48 h. Finally, cells were collected for quantitative real-time polymerase chain reaction (qRT-PCR) analysis and cell functional assays.

### 4.8. Luciferase Reporter Assay

The 3′- untranslated region (UTR) region of GLTP combined with miR-196a (WT) and corresponding mutant sites (MUT) were synthesized by Tsingke company, (Beijing, China). Then, the fragments were annealed and cloned into pmiRGLO luciferase vector (Addgene, Watertown, MA, USA). The GLTP-WT or GLTP-Mut plasmids were co-transfected with miR-196a mimic in 293T cells for 48 h, and the 2 kb upstream sequences miR-196a transcription start site were cloned into PGL3 vector. Then, the luciferase reporter plasmids were co-transfected with NRF2 overexpression vector or control vector in 293T cells for 48 h. These luciferase activities were determined using the Dual Luciferase Reporter Assay kit (Promega, Madison, WI, USA) according to the manufacturer’s instruction. The relative luciferase activities were normalized to those of Renilla.

### 4.9. Tumor Growth Assay In Vivo

Four-week-old female nude mice were purchased from Vital River Laboratory Animal Center (Beijing, China) and maintained in AAALAC-accredited special pathogen-free conditions. All experiments were conducted with the standard operating procedures approved by the University Committee of Zhengzhou university. The specific cells (3 × 10^6^) were injected into the lower armpits of nude mice, and after 2 weeks of injection, gefitinib was administered every two days by oral gavage at 25 mg/kg. Tumor sizes were monitored by vernier caliper every week and the tumor volumes were calculated as 0.5 × (Length × Width ^2^). All mice were sacrificed at the end of experiments according to the institutional guidelines and regulations.

### 4.10. Immunohistochemistry (IHC) Assay

Fresh tumor tissues from the mice were fixed with formalin, then embedded in paraffin. Tumor tissues were stained by immunohistochemical test kit (ZSGB-BIO, Beijing, China). Slices were incubated at 4 °C overnight with primary antibodies against Ki67 (1:50, Abway, Shanghai, china), BAX (1:100, Abway, Shanghai, China) or BCL-2 (1:50, Abway, Shanghai, china), stained with biotin-labeled secondary antibody (ZSGB-BIO, Beijing, China) for 2 h at room temperature, then the immunoreactivity was detected by incubating with DAB. The stained slides were counterstained with hematoxylin for cell nucleus.

### 4.11. Statistical Analysis

All data were presented as mean ± S.D. and at least three replicated separate experiments were performed for each group, except those specifically indicated. Statistical analysis was determined using Student’s *t*-test with GraphPad Prism 8 Software. The difference was considered statistically significant at *p* < 0.05.

## Figures and Tables

**Figure 1 ijms-23-01785-f001:**
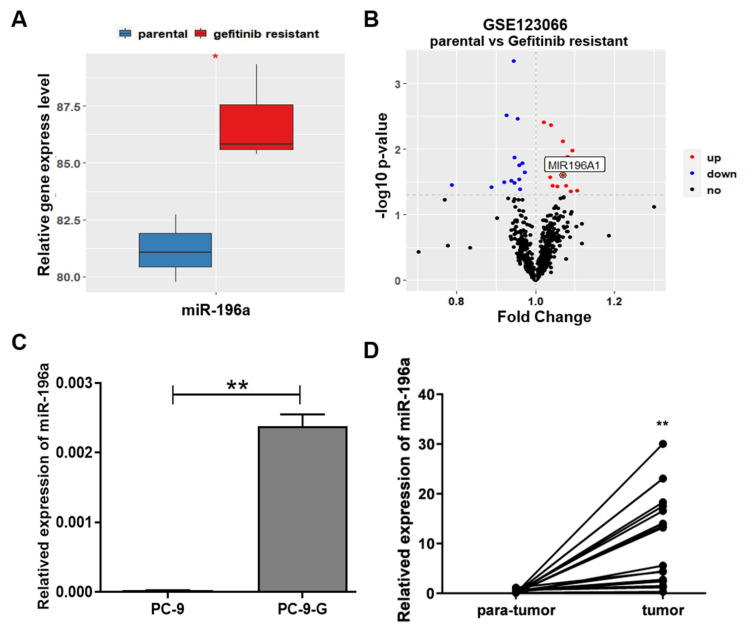
miR-196a levels were upregulated in the gefitinib-resistant lung cancer cells and were correlated with lung cancer patient survival. (**A**) The relative gene expression levels of miR-196a were examined in the parental and gefitinib-resistant lung cancer cells in GEO database. (**B**) The gene expression levels were analyzed in GSE123066 database by Volcano plot, red dots represent high expression levels in the gefitinib-resistant group compared to parental group. (**C**) The miR-196a mRNA levels were measured in PC-9 and PC-9-G cells by qRT-PCR. (**D**) The two connected dots represent the miR-196a levels in the tumors and corresponding adjacent tissues from the same patients (n = 17). Data represent the mean ± S.D. of 3 independent experiments where * indicates significant difference at *p* < 0.05, ** indicates at *p* < 0.01.

**Figure 2 ijms-23-01785-f002:**
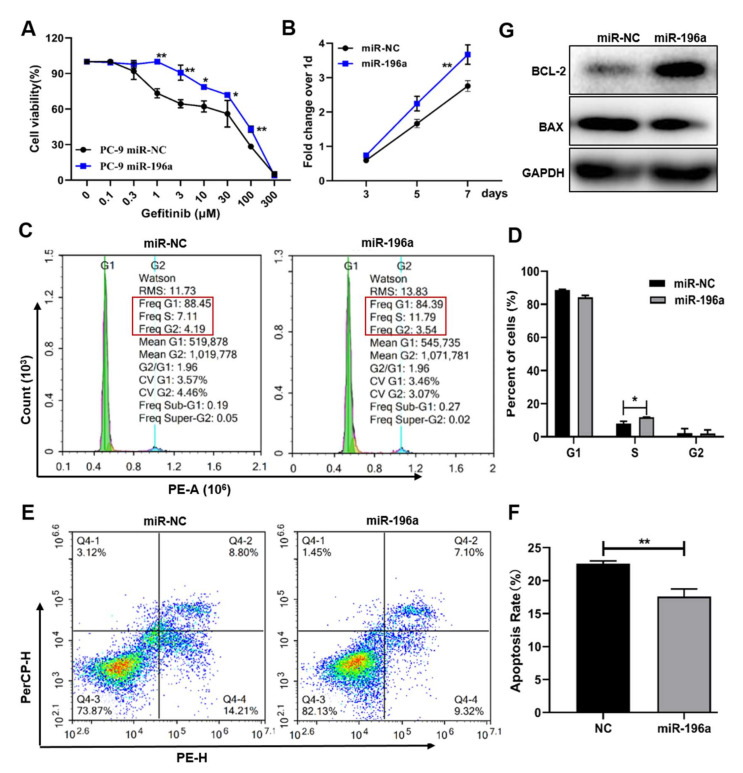
Forced expression of miR-196a-induced gefitinib resistance by promoting cell proliferation and inhibiting apoptosis. We constructed a stable overexpression cell line of miR-196a in PC-9 cells and treated them with 1 μM gefitinib for 48 h. (**A**) The cell viability following a 48-h exposure to different concentrations of gefitinib was determined in PC-9-NC and PC-9-miR-196a cells by Cell Counting kit-8 (CCK8) assay. (**B**) Cell proliferation ability was measured using the CCK8 assay as described in the methods. (**C**) Cell cycle was detected by flow cytometry analysis in PC-9 cells treated with gefitinib. (**D**) Quantitative analysis was investigated with different stages of cell cycles from three independent experiments. (**E**) Cell apoptosis was performed with flow cytometry according to the manufacturer’s protocol. (**F**) Percentage of cell apoptosis cells as shown in (**E**). (**G**) The protein expression levels of apoptosis relative protein BCL-2 and BAX by Western blotting. Data represent the mean ± S.D. of 3 independent experiments where * indicates significant difference at *p* <0.05, ** at *p* <0.01.

**Figure 3 ijms-23-01785-f003:**
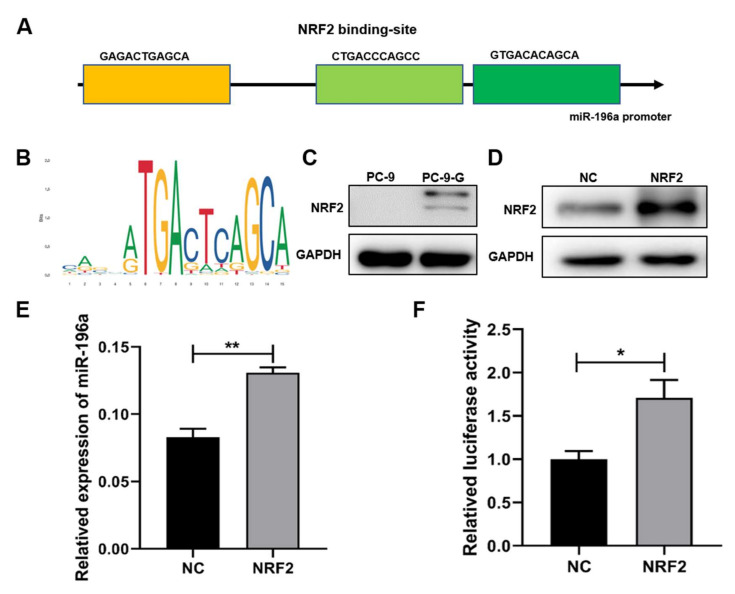
NRF2 upregulated miR-196a expression at transcriptional level. (**A**) The position and sequence of NRF2 binding to miR-196a promoter were predicted using Jasper database. (**B**) Transcriptional regulatory motif of NRF2 in gene promoter region. (**C**) The protein expression levels of NRF2 were detected in PC-9 and PC-9-G cells by Western blot. (**D**) The protein expression of NRF2 in NRF2 overexpression cells by Western blot. (**E**) The expression levels of miR-196a were tested in NRF2 overexpression cells by qRT-PCR. (**F**) The 2 kb upstream sequence of miR-196a transcription start site was cloned into PGL3 vector, which co-transfected with NRF2 overexpression vector in 293T cells. The potential NRF2 binding regions in miR-196a promoter were tested by the luciferase activities. Data represent the mean ± S.D. of 3 independent experiments where * indicates significant difference at *p* < 0.05, ** at *p* < 0.01.

**Figure 4 ijms-23-01785-f004:**
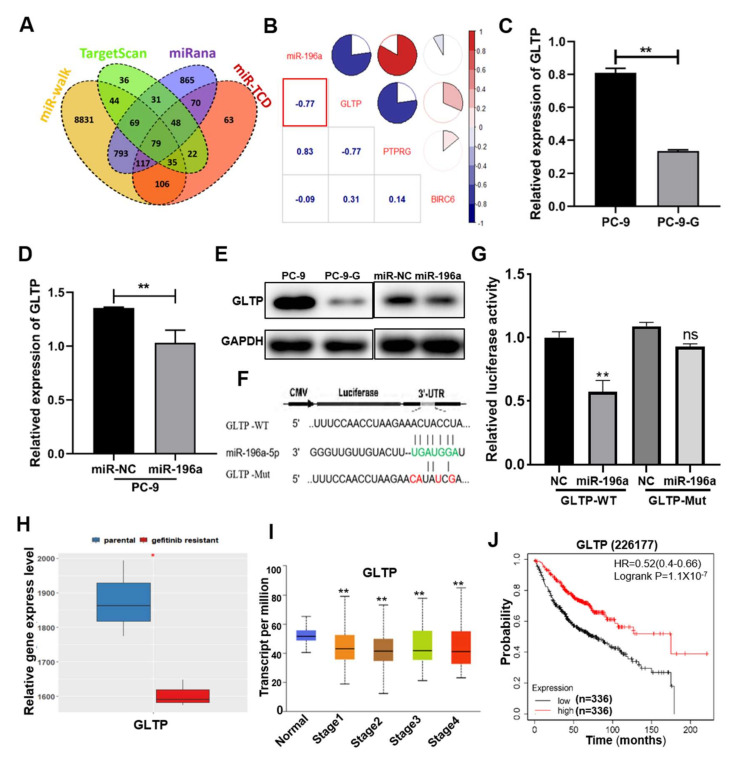
GLTP was a direct and functional target of miR-196a and GLTP levels were correlated with NSCLC development. (**A**) Venn diagram showed the number of predicted target genes of miR-196a based on miR-walk, TargetScan, miRana and miR-TCD databases. (**B**) Correlation analysis of miR-196a and its predicted top target genes in database. (**C**) The mRNA expression levels of GLTP in PC-9 and gefitinib-resistant PC-9-G cells. (**D**) The mRNA expression levels of GLTP were analyzed in miR-196a overexpression cells using qRT-PCR. (**E**) The protein levels of GLTP were detected in both gefitinib resistance cells and miR-196a overexpression cells by Western blotting. (**F**) Predicted miR-196a seed-matching sequence and designed mutant sequences in the 3′-UTR of GLTP. (**G**) The potential miR-196a binding region in the 3′UTR of GLTP was tested using luciferase reporter assay. (**H**) The relative expression levels of GLTP were analyzed in the parental and gefitinib-resistant lung cancer cells in GEO database. (**I**) GLTP expression in patients at various stages of lung cancer were examined in TCGA database (normal n = 59, Stage1 n = 277, Stage2 n = 125, Stage3 n = 85, Stage4 n = 28). (**J**) Kaplan–Meier survival curves and GLTP expression levels were analyzed using data from the TCGA database (low expression = 336, high expression = 336). Data represent the mean ± S.D. of 3 independent experiments where * indicates significant difference at *p* < 0.05 ** at *p* < 0.01.

**Figure 5 ijms-23-01785-f005:**
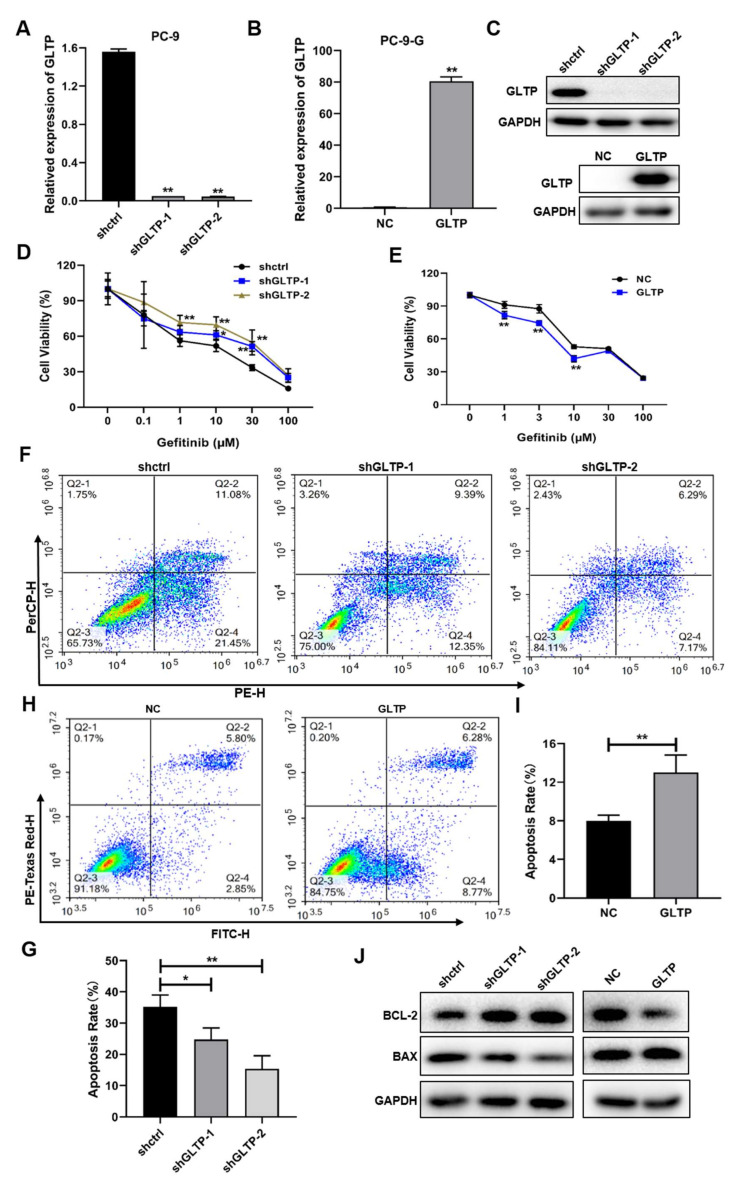
GLTP knockdown increased lung cancer cell resistance to gefitinib treatment through the inhibition of cell apoptosis. (**A**–**C**) GLTP knockdown and overexpressing stable cell lines were constructed by lentivirus infection, then selection; the expression levels of GLTP were analyzed by qRT-PCR (**A**,**B**) and Western blotting (**C**). (**D**,**E**) The cell viabilities were analyzed in cells with GLTP knockdown (**D**) or overexpression (**E**) and treated with different concentrations of gefitinib for 48 h using CCK8 assay. (**F**) Flow cytometry was used to measure cell apoptosis in GLTP knockdown cells after gefitinib (PC-9, 1μM) treatment for 48 h. (**G**) Apoptosis rates of cells as in (**F**) were illustrated by histogram. (**H**) GLTP overexpression cells were treated with gefitinib (PC-9-G, 5 μΜ) for 48 h; then, the proportion of cell apoptosis were measured by flow cytometry. (**I**) Cell apoptosis rates were quantitatively calculated from three independent experiments. (**J**) The protein expression levels of BCL-2 and BAX were detected by Western blotting in cells with GLTP knockdown or overexpression. Data represent the mean ± S.D. of 3 independent experiments where * indicates significant difference at *p* < 0.05, ** at *p* < 0.01.

**Figure 6 ijms-23-01785-f006:**
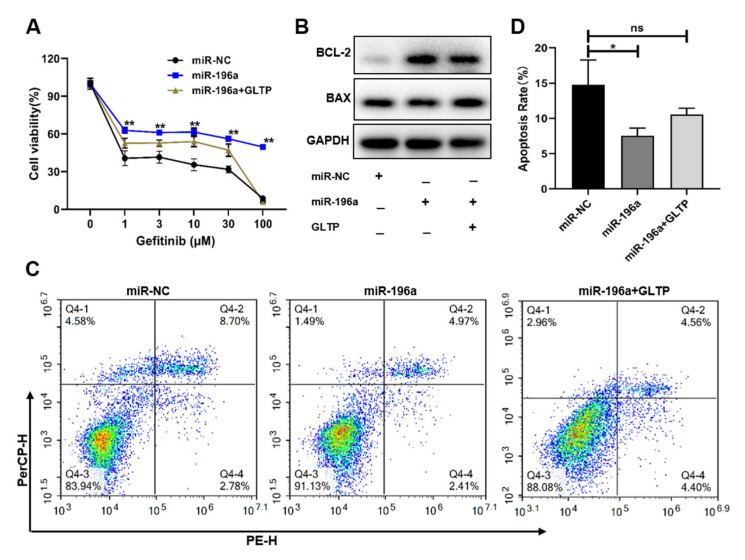
**Forced expression of GLTP partly reversed miR-196a-inducing gefitinib resistance**. PC-9 cells were transfected with GLTP overexpression lentivirus, then treated with miR-196a mimics. (**A**) Different concentrations of gefitinib were used to treat these cells for 48 h, CCK8 assay was performed for analyzing the cells viabilities. (**B**) The protein levels of apoptosis-related protein BCL-2 and BAX were measured in these cells by Western blotting. (**C**) The apoptosis levels were measured in the cells. (**D**) Quantitative analysis of apoptosis cell percentage was shown in (**C**). Data represent the mean ± S.D. of 3 independent experiments where * indicates significant difference at *p* < 0.05 ** indicates at *p* < 0.01, and ns denotes no significantly.

**Figure 7 ijms-23-01785-f007:**
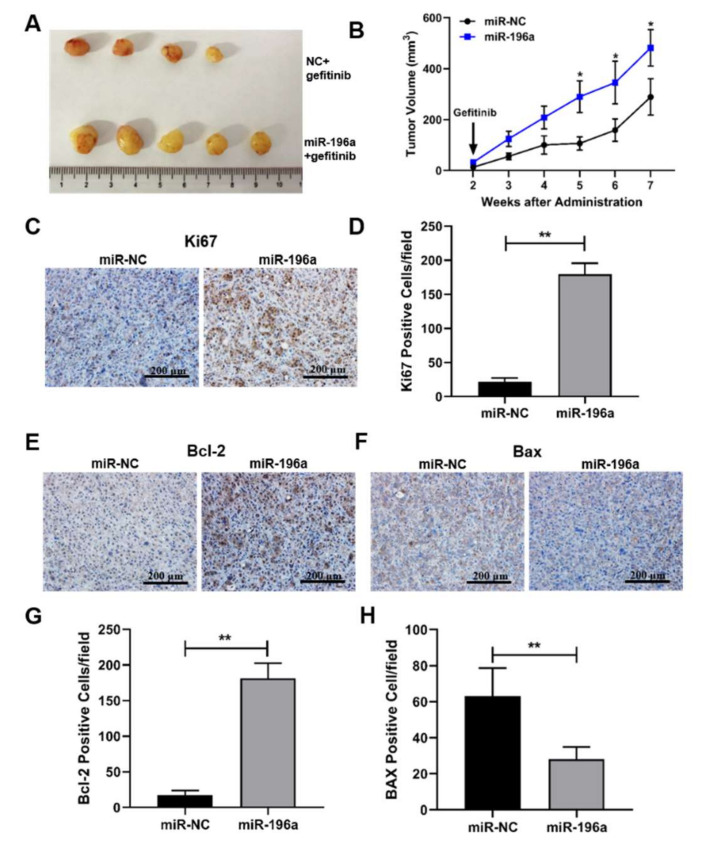
**miR-196a-forced expression promoted tumor growth and inhibited cell apoptosis by treatment gefitinib in vivo**. PC-9 cells (3 million) overexpressing NC control or miR-196a were subcutaneously injected into nude mice. When the tumor volume reached 50 mm^3^, gefitinib was given by gavage every two days. (**A**) The miR-NC and miR-196a overexpression group tumors were stripped out at the end of treatment and tumor images were shown. (**B**) The tumor volumes were measured every week, and tumor growth curves were shown. (**C**,**D**) The representative images and quantitative signals of Ki67 staining were analyzed by IHC in tumor sections. (**E–H**) The representative images and quantitative results were analyzed in BCL-2 and BAX by IHC staining in tumor sections. Data represent the mean ± S.D. of 3 independent experiments where ** indicates significant difference at *p* < 0.01 * indicates at *p* < 0.05.

## Data Availability

All datasets included within the article and the public databases GEO and TCGA will be provided unpon the request(https://www.ncbi.nlm.nih.gov/geo/query/acc.cgi?acc=GSE123066).

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
