# Peer review of "miR-196a Upregulation Contributes to Gefitinib Resistance through Inhibiting GLTP Expression"

_ijms, 2022, doi:10.3390/ijms23031785_

Round 1

Reviewer 1 Report

Attached file

Author Response

Dear reviewer, we appreciate the review’s enthusiastic comments about our work and findings.

Reviewer 2 Report

I found the manuscript: “miR-196a upregulation contributes to gefitinib resistance through inhibiting GLTP expression” from bingjie liu et al. interesting; authors did a lot of work. However, there are too many major issues that prevents its publication at this stage.

Major concerns:

The paper needs an extensive editing in terms of style and english language (there are several typos and grammar mistakes).

Regarding the style:

  • The abstract is a patch work from the results and contains information such as the type of techniques or the type of statistical test used.... these infos find their place somewhere else.
  • The results are basically a listing or a short summary of what is shown in the figures. A more accurate and thorough description of the figure is much needed.
  • Paragraph organization may be revised for better clarity (i.e. including Figure 4 and Figure 5 in the same paragraph is too much).
  • Some patients’ samples data are quite convincing, why have they been relegated to supplemental figures?
  • The figures: some of them are not (or barely) readable (i.e. figure 2C, figure 4H and the percentages of the quadrants of the apoptosis assays …). All along the manuscript there is no consistency in font size of the figures, please modify.
  • Legend to Figures: the type of statistical test used should be indicated in here.

Major technical/methodological concerns:

  • Figure 2A: why the author choose to push the viability assays up to 300uM or Gefitinib? It is a non-relevant non-physiological concentration. Why do they treat PC9 cells for 24h and not 72h? (which is normally accepted for viability assays with TKIs). 72h treatments will allow to observe the effect of gefitinib on cell proliferation at nM range avoiding drug off-target effects.
  • Figure 2B, 2C, 2D,2E, 2F: which concentration of gefitinib has been used? DMSO controls should be included at least for some experiments.
  • Figure 3D: authors talk about “NRF2 overexpression stable cell line” when measuring by qPCR the expression level of miR-196a. What is the level of NRF2 overexpression? On which cell line it is not clear in the text.
  • Legend to Figure 3: contains the info about the NRF2-Luc reporter missing in the text.
  • In the material and methods authors talk about apoptosis assays mentioning FITC-AnnexinV kit (from BD). However ,in many experiments (Figure 2E, 5F, 6C) authors present PerCP (y axis) and PE (x axis).
  • Figure 4-5: The link miR-196a-GLTP and its feedback regulation it is quite interesting, author need to better explain it, the text is not clear. A recapitulative drawing might be helpful.
  • Figure 4I: y-axis title is missing.
  • Figure 6A: see comment figure 2A above
  • Figure 7 is missing a very important control. In figure 2B authors showed that PC9-miR196a grow faster than PC9-miR control cells. In light of that their claim in figure 6B that PC9-miR196a are more resistant to gefitinib in vivo cannot be  supported without showing vehicle control groups all along the figure. Especially because PC9-miRcontrol tumors seem to grow in presence of gefitinib

Reviewer 3 Report

Summury

The study described the involvement of miR196a in the development of acquired resistance to EGFR-TKI treatment in NSCLC. The authors identified that in gefitinib-resistant cells, increased expression of the transcription factor NRF2 enhanced miR196a expression, which in turn led to reduced GLTP expression. They demonstrated that GLTP was an important target of miR-196a in promoting resistance to gefitinib through cellular apoptosis. Therefore, they suggested that the NRF2 / miR-196a / GLTP pathway could be a novel therapeutic target for overcoming gefitinib resistance.

The research project provides new and interesting results in the field of resistance mechanisms to TKI treatment in lung cancer. However, there are some aspects to be clarified:

  • Gefitinib-resistant patients from the databases (TCGA and GSE123066) have de-novo TKI resistance. Therefore the increased expression of miR196a is not a consequence of an acquired resistance mechanism.
  • I suggest adding a figure or table showing NRF2 is the highest scoring transcription factor.
  • In figure 5G the immunoblotting analysis shows a complete absence of the expression of the GLTP protein in PC-9-G cells, which does not occur in the previous figure 4E. In the latter, in fact, the protein appears to be expressed even if at low levels.
  • The expression levels of miR-196a analyzed in xenograft tumor tissues by qRT-PCR are a consequence of subcutaneous injection of tumor cells that overexpress miR196a. Therefore, I suggest that it is superfluous to show Figure 7C in the main text. It could be inserted in the supplementary figures
  • There are inconsistencies on the timing and methods of drug administration reported in “materials and methods” section respect to paragraph 2.6.
  • In the paragraph 2.4 the sentence “The results showed that GLTP levels were the highest expression target gene of miR-196a in gefitinib resistance cells” is not very clear. If GLTP is a target of miR196a, I assume its expression is lower in gefitinib-resistant cells than in parental cells. However, supplementary Figure 2A appears to show the opposite situation.

The English language is of sufficient quality. However there are some corrections to be made in some sentences probably due to typing errors. To be reviewed:

“In this study, we investigated that whether miR-196a expression was greatly induced in gefitinib resistant cells and TKI-resistant lung cancer tissues.”

“Targeting epidermal growth factor receptor (EGFR) treatment for NSCLC, EGFR-ty-rosine kinase inhibitors (EGFR-TKIs) may significantly improve the median progression-free survival (PFS) of patients carrying specific site mutations or deletions of EGFR [4].”

“: (4) The relevance of miR-196a expression in lung cancer therapeutic resistance in vivo.”

“(H) GLTP overexpression cells were treated gefitinib for 48h, then cell apoptosis levels were measured by Flow cytometry.”

Reviewer 4 Report

Title: "miR-196a Upregulation Contributes to Gefitinib Resistance through Inhibiting GLTP Expression"

COMMENTS TO THE AUTHORS

This study aimed to investigate whether miR-196a was over-expressed in gefitinib resistant cells and TKI-resistant lung cancer tissues contributing to the resistant phenotype. The study is interesting and in the scope of the Journal, but the conclusions are too strong for the presented data. The manuscript might be considered for publication after careful major revisions.

Main comments:

While the presented data is technically well performed and analyzed, I fell it might hugely benefit if someone with expertise in clinical oncology/lung cancer specialist would critically revise it. The clinical content and statements are at times erroneous or misplaced. I recommend careful revision of all concluding results/conclusions as authors should express caution in data interpretation, especially in connection to patient survival. The fact that one marker is over- (or under) expressed in patients who survive longer is not proof of a causal effect of this marker. To be able to state that miR-196a led to a resistant phenotype, one must first exclude that the sample had a resistant EGFR mutation (missing data from the manuscript, please include for both in vitro and in vivo studies), or that some demographic or clinical factor affected resistance by performing subgroup analyses and validating the data with uni- and multi-variate statistical analyses. If these analyses can not be performed on patient samples, please state that as a limiting factor for the in vivo part of the studies.

Specific comments:

  1. Introduction

“Surgery, chemotherapy and radiotherapy are the main therapeutic methods for lung cancer treatments; however, the 5-year survival rate of lung cancer patients is still very low [2,3]. The targeted therapy is currently a new promising treatment for lung cancer patients.”

These sentences need to be updated to include immunotherapy and to state which stages of lung cancer actually benefit from targeted therapy. Also, references 2 and 3 used in this section date back to 2004. Please use most recent guidelines for lung cancer treatment (ESMO, NCCN...).

  1. Introduction

“almost all patients finally develop acquired resistance after 10-12 mouths of treatment [7]. Therefore, it is important to understand new mechanisms of EGFR-TKI-acquired resistance to explore new approach for lung cancer therapy in the future.”

Please update this section with information on primary resistance to EGFR-TKIs.

  1. Introduction

“But the role of miR-196a in TKI therapy resistance in lung cancer is not known yet”

Please briefly discuss papers that have previously explored the role of miR-196a in lung cancer in general, such as (the reviewer is not a coauthor):

https://www.ncbi.nlm.nih.gov/pmc/articles/PMC3503718/

https://academic.oup.com/jb/article-abstract/166/4/323/5498562?redirectedFrom=fulltext

  1. Materials and Methods

“4.1. Human lung cancer specimens

The patients did not receive surgical treat-ment, radiotherapy, chemotherapy or immunization inhibitory therapy before surgery.”

Please amend “immunization inhibitory therapy” to immunotherapy (or what was intended to be stated).

  1. Materials and Methods

“4.1. Human lung cancer specimens

In this study, the human lung cancer tumor tissues and adjacent normal tissues of lung cancer patients were obtained from the tissue bank of the Affiliated Cancer Hospital of Zhengzhou University”

Please define the inclusion criteria for the study, tumor stage, etc. Please define how OS, PFS were determined.

  1. Results

2.1.

“tumor tissues from gefitinib treatment-resistant patients than those from gefitinib sensitive patients”

Please clearly define how gefitinib resistance was defined (after how many months of treatments, whether it was primary or secondary resistance etc.)

  1. Results

“Taken together, these results indicated that higher expression levels of miR-196a in lung tumor tissues were associated with gefitinib resistance and poor survival outcome, which indicates that miR-196a is a potential therapeutic target to overcome gefitinib re-sistance.”

Please amend and reassess this conclusion – the fact that higher expression levels of miR-196a coincided with shorter survival is not a direct association nor a proof of causality.

  1. Results

“All these results suggested that miR-196a played a regulatory role in the occurrence and development of lung cancer as well as gefitinib resistance.

Please amend and reassess this conclusion – the performed experiment did not prove a direct regulatory role of miR-196a neither in the occurrence and development of lung cancer, nor in gefitinib resistance. Use more cautioned expressions as “might contribute to” or similar.

  1. Discussion

As for conclusions, please use more cautioned expressions when discussing the obtained results. The manuscript would also benefit from a short section elaborating on the possible ways a specific therapy would be designed and delivered targeting the proposed new marker.

Round 2

Reviewer 2 Report

  • Minor English changes required.
  • The quality of some figures can be improved; for example (but not only) the apoptosis plots: quadrants title can be removed, percentage of + cells per quadrant should appear only once. 

Nonetheless the authors improved remarkably the quality of the manuscript, especially the text. Therefore I would recommend the manuscript for publication after minor revision.

Author Response

Comment-1 Minor English changes required.

Response: We have carefully revised the manuscript.

Comment-2 The quality of some figures can be improved; for example (but not only) the apoptosis plots: quadrants title can be removed, percentage of + cells per quadrant should appear only once.

Response: We have revised the figures, especially the apoptosis plots in revised manuscript.

Reviewer 3 Report

The authors abundantly revised the text of the manuscript and followed major part of Reviewer's suggestions.

Some concerns yet:

  1. A figure showing miR-196a upregulation after transfection is not present in the paper. It's very important.
  2. in Tranfection paragraph in M&M the authors stated the concentration of miR-mimic used 100microM, in my opinion is a very huge amount. Too much. Very distant from what it can detected in patients I supposed.
  3. About Fig 5C, it could be more correct adding an image of long exposure of WB about NRF2, so the low level in PC 9 G can be showed.

Author Response

We thank you  for your further comments and suggestions. Please see the attachment for the reply to the comments.

Reviewer 4 Report

I thank the authors for the efforts to review the manuscript. I have only 2 minor comments to be resolved (obligatory before acceptance):

1. Page 1 of the revised manuscript: Please revise this sentence “Epidermal growth factor receptor (EGFR) gene mutation occurs in about 10% of NSCLC tumors.” to “Epidermal growth factor receptor (EGFR) gene mutation occurs in about 10-50% of NSCLC tumors depending on the population.” For referencing you may use some of the suggested references or any similar:

https://pubmed.ncbi.nlm.nih.gov/30941946/

https://www.ncbi.nlm.nih.gov/pmc/articles/PMC4633915/

2. Comment-6: 2.1.

Please clearly state in the text that Figure 1D does not represent an analysis of patient survival that has any connection to gefitinib resistance. If this plot is only to present that LUAD patients survive differently when miR-196a levels are taken into account please reconsider to delete it, or place it in some other part of your manuscript, presenting general data on the expression of miR-196a in LUAD patients. Figure 1D does not belong to this graph section at all if it has no connection to gefitinib resistance.

Author Response

We thank you  for your further comments and suggestions. Please see the attachment for the reply to the suggestions.
